# Terbium(III) as a Fluorescent Probe for Molecular Detection of Ascorbic Acid

## Natalia Selivanova * and Yuriy Galyametdinov

Department of Physical and Colloid Chemistry, Kazan National Research Technological University, Kazan, Tatarstan 420111, Russia; yugal2002@mail.ru
* Correspondence: natsel@mail.ru; Tel.: +7-843-231-4177

**Abstract:** Fluorescence analysis is a simple and a highly sensitive method for detection of small amounts of biologically active substances. In this study, a complexation of terbium(III) chelates with 1,10-phenanthroline and ascorbic acid (AA) and luminescent properties of complexes were investigated. The influence of pH and solubilization of complexes by micellar solutions of nonionic, cationic, and anionic surfactants on fluorescence was studied. The quenching effect of terbium ion fluorescence was detected upon an introduction of ascorbic acid. The quenching effect of the complex with mixed ligands Tb(1,10-phenanthroline)-AA allows for the detection of ascorbic acid with the limit of $7.4 \times 10^{-5}$ mol·L$^{-1}$.

**Keywords:** fluorescence; terbium complex; ascorbic acid; quenching effect; micelles

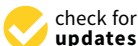



## 1. Introduction

The fluorescence analysis is a method that utilizes sensitized luminescence of lanthanide complexes and is a highly sensitive technique broadly used in analytical chemistry and scientific research in biology and biomedicine [1,2]. This method is suitable for the detection of compounds that are capable of forming complexes with lanthanide ions Ln(III). In such complexes, an analyte acts as either a sensitizer or a quencher. Due to the unique photophysical properties of Ln$^{3+}$ complexes, such as a narrow emission band and a high quantum yield of luminescence, the respective optical probes are widely used as highly sensitive analytical sensors for visualizing various lesions in cells and tissues, drug delivery monitoring, and clinical analysis [3,4]. Recent studies report the applications of lanthanide-doped nanoparticles as biosensors for COVID-19 detection [5].

In the aspect of molecular recognition of various drugs, the development of simple, rapid, and highly sensitive methods for their detection is highly demanded. Optical probes based on lanthanide complexes are of considerable interest, providing the required accuracy, sensitivity, and rapidity of determination. Using sensitized luminescence of lanthanides, drugs of various classes are determined, the molecules of which are capable of complexation with Ln$^{3+}$ ions: derivatives of aminobenzoic acid [6]; non-steroidal anti-inflammatory drugs [7,8]; antibiotics of the tetracycline series [9].

Ascorbic acid, also known as vitamin C, performs the biological functions of a reducing agent and a coenzyme of metabolic processes. It is one of the main substances in the human diet which is necessary for the normal functioning of connective bone tissue. It plays an important role in such biological processes as the capture of free radicals, synthesis of collagen protein and a number of hormones including anti-stress, helps maintain hemostasis, improves the body's ability to absorb calcium, removes toxins, and regulates metabolism [10]. Various analytical methods are used for the detection of ascorbic acid, including spectrophotometric, chromatographic, calorimetric, and electrochemical ones [10–13]. Despite the fact that the listed methods are sensitive and very specific, there are still some drawbacks. These methods are intricate and time-consuming, and some of them usually required specialized and expensive instruments. Compared to other detection

methods, fluorescent methods get more attention due to their good reproducibility and high sensitivity. Recently, the fluorescent method has been developed for AA sensing based on nitrogen-doped carbon dots [14].

Terbium(III) as a commonly used lanthanide ion in analytical chemistry. It is characterized by a large radius, versatile coordination geometry, and a high affinity to ligands containing oxygen or hybrid oxygen-nitrogen atoms. As summarized in the review [15,16], the $Tb^{3+}$ sensitized fluorescence method is broadly used for the determination of pharmaceuticals.

The detection of ascorbic acid based on the sensitized luminescence of terbium(III) was reported in [17,18]. The authors used the complex of terbium(III) nitrate with hydroxymethyl in methanol and the complex of terbium(III) chloride with acetylacetone in aqueous medium. Detection of the limits of ascorbic acid in the first case were $1.2 \times 10^{-7}$ mol·$L^{-1}$, in the second $8 \times 10^{-3}$ mol·$L^{-1}$.

The aim of this work was to study the complexation between ascorbic acid and $Tb^{3+}$ ions in the presence of the second ligand 1,10-phenanthroline and the micelles of different surfactants. The optimal terbium fluorescence conditions for ascorbic acid detection were determined.

## 2. Materials and Methods

### 2.1. Materials

Terbium nitrate pentahydrate $Tb(NO_3)_3 5H_2O$, 1,10-phenanthroline $C_{12}H_8N_2$(Phen), decaethylene glycol monododecyl ether $C_{12}H_{25}O(CH_2CH_2O)_{10}H(C_{12}EO_{10})$, sodium dodecyl sulfate $CH_3(CH_2)_{10}CH_2OSO_3Na$(SDS), hexadecyltrimethylammonium bromide $CH_3(CH_2)_{15}NBrCH_3$ (HTAB) are commercial products from Aldrich and are used without additional processing. Ascorbic acid $C_6H_8O_6$(AA) from Tatfarm Company. Solutions of all basic and auxiliary chemical reagents were prepared using double-distilled water. Acetate-ammonia buffer solution was prepared from 2 M $CH_3COOH$ and $NH_3$ solutions.

### 2.2. Preparation of Samples

Synthesis of the Tb(Phen) complex is carried out in an aqueous medium containing $1.6 \times 10^{-3}$ M of surfactant based on a 1:1 molar ratio of the components. The sample of Phen was initially dissolved in 0.1 mL of ethanol. The solution was further stirred at room temperature for 2 h.

The concentration of ascorbic acid in stock standard aqueous solutions is $4 \times 10^{-3}$ M.

To plot a calibration graph, 1 mL of a buffer solution with pH = 7.0 was added to the test tube. The following components were then added: 0.5 mL of $Tb^{3+}$ $1 \times 10^{-3}$ M salt solution, 0.5 mL of Phen $1\times10^{-3}$ M, 0.5 mL of $C_{12}EO_{10}$ $1.6 \times 10^{-3}$ M. Various additives of AA are dissolved in 5 mL of the buffer solution to provide the concentration range of $1 \times 10^{-6}$ M–$1 \times 10^{-4}$ M.

### 2.3. Detection of Ascorbic Acid in Tablets

Five tablets containing 100 mg of ascorbic acid were ground into a fine powder. The powder was then divided into five equal parts. These parts were dissolved in water and filtered through a quantitative filter paper. The filtrates were collected and diluted with water to prepare solutions with different concentrations of ascorbic acid.

### 2.4. UV/Vis

Electronic absorption spectra were recorded on a Lambda 35 Uv-Vis Spectrometer (PerkinElmer Ltd., Buckinghamshire, UK). We used quartz cells with an optical path length of 1 cm.

### 2.5. Fluorescence

Fluorescence spectra were recorded on the Cary Eclipse (Varian Australia Pty Ltd., Mulgrave, Australia) spectrofluorimeter. The measurements were carried out in a quartz cells with a thickness of 1 cm; the signal was recorded at the angle of 90° to the excitation

light. The filters were used in the automatic mode. The parameters of the excitation and emission slits were 20 nm and 5 nm, respectively. The emission spectra were obtained at the excitation wavelength of 303 nm. The excitation spectra were recorded by detecting the emission intensity of $Tb^{3+}$ ions at 544 nm. For the emission lifetime, the fluorescence intensities at $\lambda_{max}$ = 544 nm were recorded at different delay times. The experimental curves were exponentially approximated by Origin software.

### 2.6. DLS

To measure the sizes of micellar aggregates, the method of dynamic light scattering was used. The studies were carried out on a Zetasizer Nano ZS apparatus with a helium–neon laser (633 nm, 4 mW) from (Malvern Instruments Ltd., Worcestershire, UK). Before measurement, the solutions were filtered through a Millipore hydrophilic filter in the Millex HV Filter Unit with 0.45 µm pores. The light scattering angle was 173°. The experiment was carried out at the temperature of 25 °C. The data on the sizes of micelles were taken from the Zetasizer Nano ZS software report.

### 2.7. pH

The pH values of the solutions were measured by a Knick pH meter. The pH was varied using 1.0 M HCl and 0.1 M NaOH solutions.

## 3. Results and Discussion

### 3.1. Complexation of Tb(III) Ions with 1,10-Phenanthroline and Ascorbic Acid

The efficiency of energy transfer and the intensity of sensitized luminescence depends on a number of factors, such as the nature of the ligand, the type of the lanthanide ion, solvent, pH of the medium, and the difference in energies between the triplet level of the ligand and the emitting level of the lanthanide. The fluorescence of Tb(III) ions occurs only from certain resonance levels. The energy of the triplet level of Phen is 22,075 $cm^{-1}$; accordingly, this value is higher than the energy of the resonance level of the terbium ion at 20,950 $cm^{-1}$ [15]. This makes possible the transfer of excitation energy from the organic ligand molecule to the terbium ion, which contributes to sensitized fluorescence.

Spectrophotometric studies were carried out to assess the interactions of the Tb(Phen) complex with ascorbic acid.

As can be seen from Figure 1, the hypsochromic shift from 267 nm to 264 nm is observed in the presence of ascorbic acid. These data indicate a possible complexation process via interactions between lone oxygen pairs in ascorbic acid molecules and vacant terbium ion orbitals.

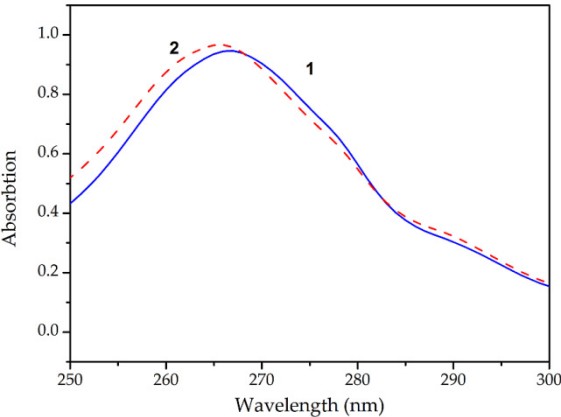

**Figure 1.** Absorption spectra of complexes **1**. Tb(Phen) and **2**. Tb(Phen)-AA.

Figure 2 demonstrates the excitation and emission spectra of the Tb(Phen) and Tb(Phen)-AA complexes. The excitation at the 303 nm wavelength leads to the formation of typical $Tb^{3+}$ emission peaks at 489, 544, 585, and 620 nm. These peaks correspond to the

following radioactive transitions of the Tb ions: 1. $^5D_4–^7F_6$, 2. $^5D_4–^7F_5$, 3. $^5D_4–^7F_4$, and 4. $^5D_4–^7F_3$, respectively, and thus indicate an intramolecular energy transfer from Phen to $Tb^{3+}$. Fluorescence at the wavelength $\lambda_{max}$ = 544 nm, which corresponds to the $^5D_4–^7F_5$ transition, results in a green emission from the samples.

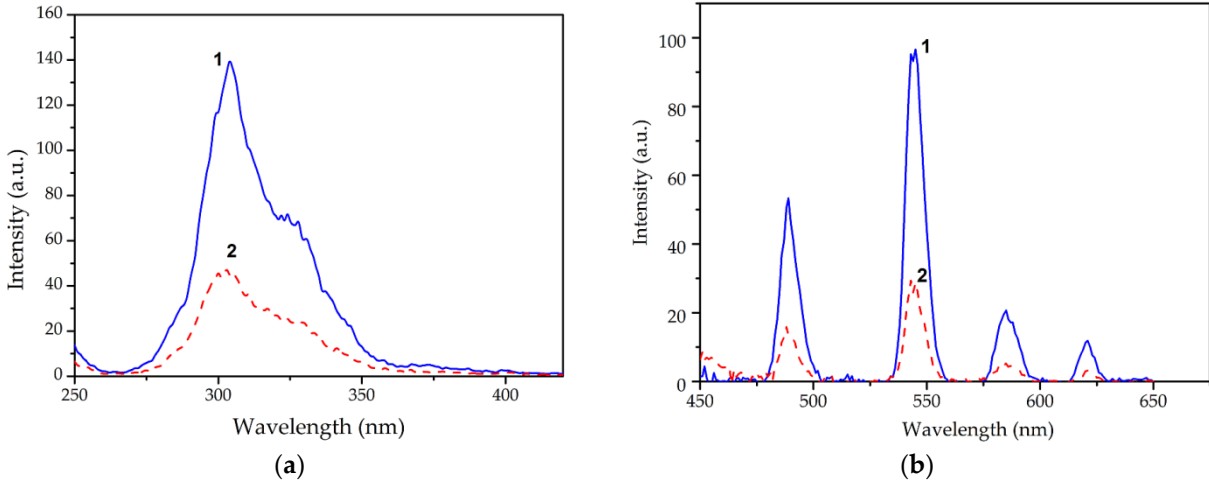

**Figure 2.** (**a**) Excitation spectra; (**b**) Fluorescence spectra of systems: **1.** Tb(Phen), **2.** Tb(Phen)-AA, $C_{AA}$ = $1.4 \times 10^{-4}$ mol·L$^{-1}$.

In comparison with the Tb(Phen) system, the fluorescence spectra of the Tb(Phen)-AA system shows a significant decrease in intensity. The effect of fluorescence quenching is observed for the Tb (III) ions. A three-fold decrease in fluorescence intensity is detected if the concentration of the added ascorbic acid is C = $1.4 \times 10^{-4}$ mol·L$^{-1}$.

### 3.2. The Effect of Surfactants on the Fluorescence of Tb(III) Ions

A well-known approach to increase the intensity of an analytical signal is the use of micelles, microemulsions, or nanoparticles in the fluorescence analysis [15]. As reported in [9], solubilization of lanthanide complexes in nanosized surfactant micelles is accompanied by an additional increase in the sensitized fluorescence intensity. This effect can be explained by an accumulation of complexes inside micelles, a convergence of reacting components, stabilization of complexes, and shielding of the luminescent particles from extraneous quenchers. On the other hand, the Tb(Phen) complex is poorly soluble in water. Micellar systems are, therefore, expected to enhance the solubility of complexes and increase the stability or their analytical signal. These effects depend on the nature of surfactants. In this regard, we studied the effect of nonionic, anionic, and cationic surfactants on the properties of complexes.

The characteristics of surfactants used in this work are summarized in Table 1. The concentrations of surfactants in solutions were higher than the critical micelle concentrations (CMC).

**Table 1.** Surfactant characteristics.

| Surfactant | Working Concentration $C_{surf}$, mol·L$^{-1}$ | CMC, mol·L$^{-1}$ | d, nm | Tb(Phen)-AA Complex Diameter, nm |
|---|---|---|---|---|
| $C_{12}EO_{10}$ | $1.6 \times 10^{-3}$ | $9.1 \times 10^{-5}$ [19] | $3 \pm 0.92$ | $7.5 \pm 1.60$ |
| SDS | $8.3 \times 10^{-2}$ | $8.1 \times 10^{-3}$ [20] | $5 \pm 1.07$ | $5.5 \pm 1.12$ |
| HTAB | $3.0 \times 10^{-2}$ | $9.1 \times 10^{-4}$ [20] | $6 \pm 1.27$ | $1.7 \pm 0.32$ |

The effect of the Tb(Phen)-AA complex additives on the size of micellar aggregates depends on the type of surfactant (Figure 3). Introduction of the Tb(Phen)-AA complex into HTAB micellar solutions reduces the size of micelles from 6 nm to 1.7 nm, indicating that no solubilization of added complexes occur in this system. The sizes of SDS and C12EO10

aggregates in the presence of this complex are 5.5 nm and 7.5 nm, respectively. In the case of an anionic surfactant (SDS), the size of micelles is almost insensitive to additions of the Tb(Phen)-AA complex. The micelles of a nonionic surfactant (C12EO10), however, grow from 3 nm to 7.5 nm. An increase in the size of C12EO10 micelles may indicate that they solubilize the Tb(Phen)-AA complex.

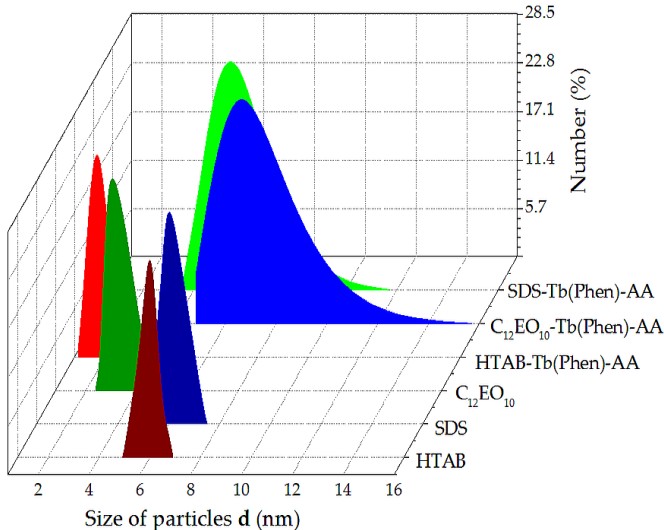

**Figure 3.** Particle size distribution curves: HTAB (⬤); SDS (⬤); C12EO10 (⬤);Tb(Phen)-AA-HTAB (⬤);Tb(Phen)-AA-SDS (⬤); Tb(Phen)-AA-C12EO10 (⬤).

To characterize the effect of surfactant micelles on the luminescent properties of the Tb(Phen)-AA complex, its fluorescence spectra (Figure S1) and lifetime (Figure 4) were analyzed. The ratio of the intensities of the transitions $I(^5D_4-^7F_5)/I(^5D_4-^7F_6)$ may be a criterion for assessing the luminescence monochromaticity of the terbium(III) complexes [19]. We therefore estimated the fluorescence efficiency of these complexes as the ratio of the main energy transition intensities $\eta = I(^5D_4-^7F_5)/I(^5D_4-^7F_6)$ at the wavelengths of $\lambda = 544$ nm and $\lambda = 489$ nm.

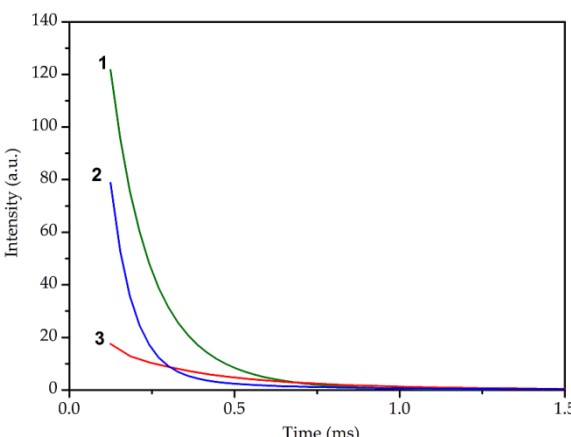

**Figure 4.** Fluorescence decay curves of the Tb(Phen)-AA complexes solubilized in the surfactant solutions: **1**. $C_{12}EO_{10}$; **2**. SDS; **3**. HTAB.

The fluorimetric data show that the fluorescence efficiency $\eta$ of the Tb(Phen)-AA complexes in the micellar solutions of surfactants does not change significantly (Table 2).

Figure 4, however, reveals the effect of surfactants on the luminescence lifetimes. It should be noted that the luminescence decay of the Tb(Phen)-AA complexes in micellar solutions is approximated by a monoexponential curve fitting.

**Table 2.** The lifetime and the fluorescence efficiency of the Tb(Phen)-AA complex in microheterogeneous media ($\lambda$ = 545 nm, transition $^5D_4 \rightarrow {}^7F_5$).

| Surfactant | Water | $C_{12}EO_{10}$ | SDS | HTAB |
|---|---|---|---|---|
| $\tau$, $\mu$s | $160 \pm 16$ | $194 \pm 19$ | $40 \pm 4$ | $187 \pm 19$ |
| $\eta$ | $1.83 \pm 0.09$ | $1.98 \pm 0.1$ | $1.98 \pm 0.1$ | $1.91 \pm 0.09$ |

Table 2 represents luminescent characteristics of the Tb(Phen)-AA complex solubilized in surfactant solutions.

As we can see from Table 2, the presence of $C_{12}EO_{10}$ micelles increases the lifetime of complexes. These data are consistent with our earlier works, which characterized interactions between lanthanide ions with nonionic surfactants [21,22]. The lifetime of the Tb(Phen)-AA complex is longer in the $C_{12}EO_{10}$ solution. According to the concepts developed in our studies, these effects can be caused by interactions between lone electron pairs in oxygen atoms of surfactant oxyethylated groups with the vacant orbitals of the $Tb^{3+}$ ions, so these ligands fill the coordination spheres of the lanthanide metal ions. A longer lifetime can be therefore associated with the solubilization of complexes in micellar solutions and a reduction of radiationless losses, so we used a nonionic surfactant $C_{12}EO_{10}$ for further studies.

### 3.3. The Effect of pH

The intensity of sensitized fluorescence of the Ln(III) complexes depends significantly on the pH of medium. This effect is associated with the acid-base equilibria of a ligand and complex as well as the hydrolysis of lanthanide ions [8,23]. Considering this, we studied the influence of pH of medium on the fluorescence efficiency of the complexes.

As can be seen from Figure 5, the fluorescence parameter $\eta$ of the analyzed complexes first increases with the increase in pH, reaching a maximum, and then decreases. For the Tb (Phen) complex, a small plateau is observed in the pH range 5.4–7.2; for the Tb(Phen)-Asc complex, the maximum value of the parameter $\eta$ is observed at pH 7–8. An increase in pH leads to a sharp decrease in the fluorescence intensity. This is probably due to decomposition of Tb(Phen) complexes at pH > 7.2 and Tb(Phen)-AA) complexes at pH > 8 with the resulting formation of terbium hydroxide $Tb(OH)_3$. A low intensity of fluorescence of these complexes under study in the acidic region indicates that such conditions do not favor formation of complexes. The Tb(Phen)-AA complex maintains its fluorescent properties in a wider range of pH = 2–10 and demonstrates the maximum fluorescence intensity at pH = 7–8. To eliminate the effect of adding AA on pH, we further studied the quenching effect of Tb(Phen)-AA complex at pH = 7 using acetate-ammonia buffer solution.

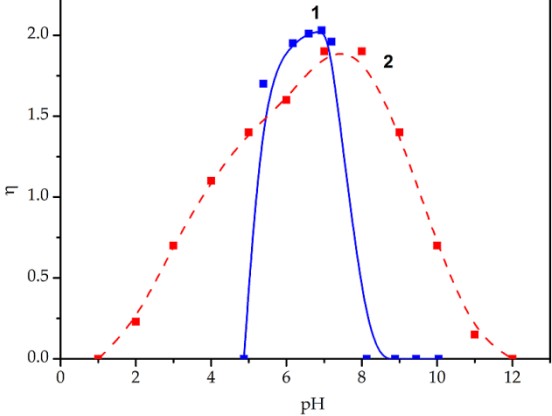

**Figure 5.** The dependence of the luminescence efficiency on the pH in $C_{12}EO_{10}$ micelles: **1**. Tb(Phen) and **2**. Tb(Phen)-AA, $C_{AA} = 1.2 \times 10^{-5}$M.

### 3.4. Quenching Effect Calibration Graph and Detection Limits of Ascorbic Acid

To optimize the conditions for detecting ascorbic acid by a fluorometric method, to increase the detection sensitivity, and to lower the detection limit, the excitation and fluorescence spectra of Tb(Phen) complexes were studied in the presence of ascorbic acid in the concentration range of $1.6 \times \cdot 10^{-6}$–$3.8 \cdot \times 10^{-4}$ M.

As we can see from the fluorescence spectra (Figure 6), the fluorescence intensity decreases upon the addition of acid.

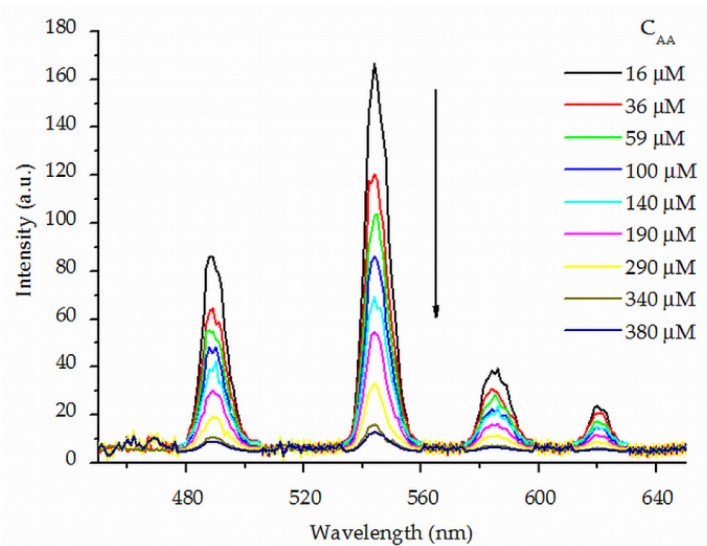

**Figure 6.** Fluorescence spectra of the Tb(Phen) complex with various additives of ascorbic acid (pH = 7).

Figure 7 illustrates a Stern–Volmer plot for fluorescence quenching of Tb(Phen) by ascorbic acid. The fluorescence intensities were measured at $\lambda = 544$ nm.

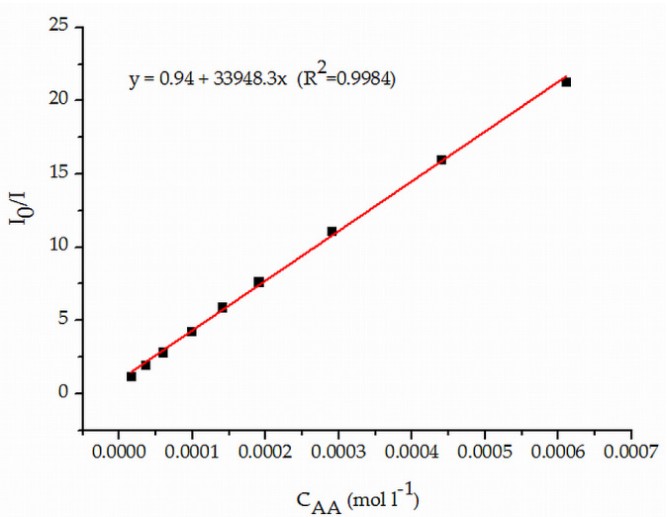

**Figure 7.** Stern–Volmer plot for Tb(Phen) quenching by ascorbic acid.

According to current concepts of fluorescence quenching types, a possible process in this case can be dynamic quenching with emissionless transfer of electron excitation energy described by a Stern–Volmer equation (Equation (1)) for a bimolecular reaction $D^* + A \rightarrow D + A^*$:

$$I_0/I = 1 + k_q \tau [A], \tag{1}$$

where $I_0$ and $I$ are fluorescence intensities with and without a quenching agent, respectively, $k_q$ is the constant of bimolecular quenching, $\tau$ is the lifetime of D* without a quenching

agent molecule, and [A] is the concentration of a quenching agent. The $\tau$ value was found from the fluorescence decay curve for Tb(Phen). The lifetimes of the individual aqueous Tb(Phen) complexes are best fitted by biexponential curves (Figure S2). We used the average value of lifetime $\tau_{av}$ = 135 $\mu$s.

The linear plot shape in the Stern–Volmer coordinates indicates that only one of the possible quenching mechanisms is in effect. The value of the bimolecular quenching constant calculated using Equation (1) is $k_q$ = 2.5 $\times$ 10$^8$ M$^{-1}\cdot$s$^{-1}$.

This quenching effect of the Tb$^{3+}$ fluorescence was used to detect ascorbic acid. Figure 8 shows the calibration plot in the "fluorescence efficiency-ascorbic acid concentration" coordinates.

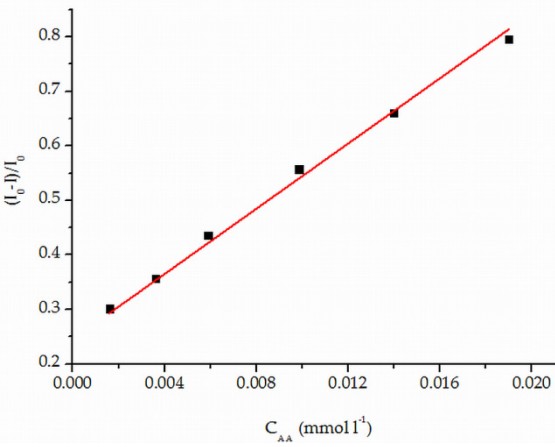

**Figure 8.** The calibration curve for ascorbic acid detection.

In the concentration range of 0.0016–0.19 mmol$\cdot$L$^{-1}$, the plot is linear. It is approximated by Equation (2):

$$y = 0.2617 + 28.936x \ (R^2 = 0.99). \tag{2}$$

According to [24], the detection limit is correctly determined by the 3$\delta$ criterion for a linear dependence of the detected value on the concentration of a substance. According to this criterion, the detection threshold of a substance is determined by Equation (3):

$$C_{min} = 3\delta_0/b, \tag{3}$$

where $\delta_0$ is the standard deviation considered for measuring the signal in the blank experiment, b is the instrumental sensitivity coefficient characterizing the change in the signal with the change in concentration, which is numerically equal to the tangent of the slope. In our case, the value $\delta_0$ = 0.71.

The detection limit of ascorbic acid $C_{min}$ = 7.4 $\times$ 10$^{-5}$ mol$\cdot$L$^{-1}$ was found using Equations (2) and (3).

To explore the practical application of Tb(Phen), the levels of ascorbic acid in tablet were measured.

As shown in Table 3, the recoveries of ascorbic acid in tablet samples were between 98.4% and 100.8%. The results demonstrate a good accuracy of ascorbic acid detection in pharmaceuticals using complex Tb(Phen) in micellar C$_{12}$EO$_{10}$ solution.

**Table 3.** Detection of ascorbic acid in tablets.

| Samples | Added, $\mu$M | Detected, $\mu$M | Recovery, % |
|---|---|---|---|
| Tablet 1 | 0 | not dedected | - |
| Tablet 2 | 50 | 52.1 $\pm$ 0.60 | 100.4 |
| Tablet 3 | 100 $\pm$ 16 | 98.4 $\pm$ 0.91 | 98.4 |
| Tablet 4 | 150 $\pm$ 0.09 | 151.2 $\pm$ 0.12 | 100.8 |

## 4. Conclusions

In this paper, we studied the complexation of terbium(III) ions with 1,10-phenanthroline and ascorbic acid and the luminescent properties of the resulting complexes. It was shown that due to the formation of a mixed-ligand complex, the fluorescence of the terbium(III) ion is quenched. Quenching of excited states can be caused by any deactivation process that results from interactions of excited molecules with ascorbic acid. The linear plot shape in the Stern–Volmer coordinates is the indication of only one quenching mechanism in effect. For the first time, the effect of dynamic quenching was revealed for the mixed ligand Tb(Phen)-AA complex. The value of the bimolecular quenching constant is $2.5 \times 10^8$ M$^{-1}$·s$^{-1}$.

The solubilizing efficiency of surfactants was estimated by the analysis of aggregation in micellar solutions with added complexes. It was revealed that the lifetime of the Tb(Phen)-AA complex depends on the type of surfactant. The longest lifetime is observed for micelles of a $C_{12}EO_{10}$ nonionic surfactant.

The efficiency of fluorescence in micellar solutions depends on the acidity of the medium. Fluorescence reaches its maximum at the optimal pH values equal to 7–8.

With optimal conditions set for the Tb(Phen) fluorescence, the quenching effect can be an effective tool for the molecular recognition of ascorbic acid with a detection limit of $7.4 \times 10^{-5}$ mol·L$^{-1}$. Facile preparation, commercial availability and high sensitivity of the proposed fluorescent probe make them prospective for detection bioactive substances.

**Supplementary Materials:** The following data are available online at https://www.mdpi.com/article/10.3390/chemosensors9060134/s1, Figure S1: Fluorescence spectra of the Tb(Phen)-AA in microheterogeneous media: Tb(Phen)-AA-C12EO10; Tb(Phen)-AA-HTAB; Tb(Phen)-AA-SDS, Figure S2: Fluorescence decay curve of Tb(Phen) in aqueous solution.

**Author Contributions:** Conceptualization, N.S. and Y.G.; methodology, N.S.; software, N.S.; validation, N.S. and Y.G.; formal analysis, N.S.; investigation, N.S.; resources, N.S.; data curation, Y.G.; writing—original draft preparation, N.S.; writing—review and editing, N.S. and Y.G..; visualization, N.S. Both authors have read and agreed to the published version of the manuscript.

**Funding:** This research was funded by Russian foundation for basic research, grant number 20-03-00620.

**Institutional Review Board Statement:** Not applicable.

**Informed Consent Statement:** Not applicable.

**Data Availability Statement:** Not applicable.

**Conflicts of Interest:** The authors declare no conflict of interest.

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
