# Peer review of "Terbium(III) as a Fluorescent Probe for Molecular Detection of Ascorbic Acid"

_chemosensors, doi:10.3390/chemosensors9060134_

Round 1
Reviewer 1 Report
Quality control of drugs in pharmaceuticals, determination of their content in biological fluids, and other objects require to development of various and highly sensitive methods for their determination. From this point of view, the presented article is relevant and has a novelty associated with both the terbium complex compound used and the assessment of the effect of surfactants on the analytical signal. There are some comments and questions about this article.
- Why do the authors call terbium not just a probe, but a microheterogeneous probe? The medium may be microheterogeneous, but not the terbium ion.
- Table 2 shows the lifetimes of the complex in micelles of various surfactants, but there is no lifetime of the complex in the absence of micelles. This does not allow us to understand the change in this parameter upon solubilization in micelles.
- The authors in Fig.2 show that terbium forms a complex with ascorbic acid. In this regard, the conclusion about the dynamic mechanism of quenching of the complex in the presence of ascorbic acid does not correspond to this. the mechanism could be proved by investigating the effect of temperature on the behavior of the Stern-Volmer dependence.
- The authors compare the detection limit of the proposed method with that of Ref.17 but say nothing about the advantages over the technique proposed in Ref.16.
- Some inaccuracies in the text of the article are marked with a different color and corrected.
Author Response
Thank you very much for your detailed and an objective review of our work and appreciated the novelty of the manuscript.
All remarks have been considered and necessary comments were added into the text.
1) You are absolutely right, microheterogeneous refers to the environment where the terbium complex is solubilized. The title has been changed.
2) The life time value for the complex in the aqueous solution was added to Table 2
Answer on 3 comment.
3) Figure 2 demonstrates the decrease in the luminescence intensity with the addition of ascorbic acid.
We make a conclusion about Stern-Volmer dynamic quenching based on the linear dependence of intensity after adding quencher molecules at different concentrations (Fig 7). This is consistent with literature data [J. Lakowicz, Principles of Fluorescence Spectroscopy (Plenum Press, New York, 1983). R. Badley, Fluorescence Spectroscopy (Plenum Press, New York, 1983)]. The temperature dependence would allow calculating the energy characteristics of the process. However, we do not have such a technical capability.
4) Authors in ref. 16 [Tan, H.; Wu, J.; Chen, Y. Terbium(III) based coordination polymer microparticles as a luminescent probe for ascorbic acid Microchim. Acta 2014, 181, 1431–1437. doi:10.1007/s00604-014-1282-x] synthesized luminescent coordination polymer Tris/Tb/DPA. Using quenching effect of luminescence Tris/Tb/DPA suspention the ascorbic acid was detected in aqueous solution with a detection limit of 200 nM. However, the authors do not provide details of obtaining this value. According to the concentration dependence given in the article, the value 200 nM cannot be achieved. For this reason, these data were not discussed in our article.

Reviewer 2 Report
This is a somehow well-organized paper with some useful information. However, some comments need to be taken into consideration for further improving the quality.
(1) A few language issues, please check the whole paper for this.
(2) You have the particle size distribution curves, but I didn't see the morphology of these particles. Some images from electron microscopy would help with that.
(3) Two Figure 2.
(4) The discussion part didn't discuss any meaningful information. You should discuss the results by comparing them to other publications.
(5) No conclusion part.
Author Response
Thank you for a positive decision on the manuscript and an objective review of our work.
1) English language was checked
2) It is rather difficult to prepare micellar solution samples for electron microscopy studies. When dried, micelles tend to aggregate. Therefore, the particles size will be different. The DLS method is reliable for evaluating micelle sizes. The main purpose of using the DLS method was to assess the change in the size of micelles upon solubilization of the complex Tb(Phen) – AA.
3) This mistake was corrected.
4,5) We combined the results and discussion parts and added a conclusion. The third part includes additional comments in comparison with the literature data.
Reviewer 3 Report
In the manuscript entitled “Terbium(III) as a fluorescent microheterogeneous probe for molecular detection of ascorbic acid” by N. Selivanova and Y. Galyametdinov, the authors report about a new method for the detection of ascrobic acid by means of fluorescence spectroscopy. In particular, a complex between Tb(III) and 1,10-phenanthroline, in combination with micelles, was proposed. I found the idea interesting. In addition, it can be useful to interested readers who work in this field. I think that, in principle, the manuscript is suitable for Chemosensors but not in the current form. Thus, I recommend publication only after major revision:
- The text needs a strong revision of the English. Some sentences are not clear and there are some mistakes (see for example, page 2, row 59 and 69, page 4 row 123);
- The method section should be rewritten better. For example, in fluorescence which wavelengths (excitation at ?) were used are not reported. How lifetimes were calculated is not written (how did the authors fit the curves?). It can be useful to use subsections (Samples preparation, UV/Vis, fluorescence, DLS);
- Figure 2 is actually Figure 1 in page 3;
- Why there are four bands in the emission spectra of the complex? Is it reported to which transitions they are due to? If yes, it should be described.
- In Table 1 there is a mistake, I guess. The authors reported that they used a concentration of monomers higher than the CMC. But from the table it seems not. For C12EO10, the concentration was lower that the CMC. For SDS, they used exactly the CMC. Instead for HTAB just a little bit higher. Please check.
- It is not clear why micelles should be used in combination with Tb-phen complex. Is it poorly soluble? Please, discuss better this point;
- Again, in Table 1, it would be useful to add a column with the d values of micelles after the complex addition. Errors on the d values should be added;
- In Figure 3, the size distribution functions for the micelles in the absence of complex should be added;
- At page 4, row 144 the authors stated that the HTAB micelles were destroyed. I think that it is not correct since they are reduced in size but still present. In addition, at row 145, it is reported that “the same weak effect is observed for SDS aggregates”. It is not clear, the same weak effect respect to what? For C12EO10 I would not say “grow slightly” since the d value after the addition of complex is more than double.
- At page 4 row 148, how it is possible to conclude form DLS that micelles of C10EO10 are the more efficient solubilizers? To state this, a partition coefficient should be measured.
- At page 5, it is not clear in which way the fluorescence efficiency is calculated. Did you use the intensities from fluorescence spectra at a particular wavelength? Please, describe better this point;
- In Table 2, the errors should be added. In addition, how the intensity decay curves were fitted is not reported (exponential, biexponential?);
- At page 5, row 165, it is reported that C12EO12 micelles increase the lifetime. But, respect to to what? Complex without micelles? Please describe better this point and lifetime of the Tb-phen-AA complex should be added as reference;
- The caption of Figure 5 is wrong. In addition, in this section please highlight that you are writing about the complex with C12EO10 micelles;
- In Figure 6, please replace volumes with concentrations in the figure legend;
- In Figure 7, at which wavelength the Stern-Volmer plot was performed. There are 4 bands in the fluorescence emission spectrum;
- Page 7, row 211, D* is not the lifetime. In row 212, the concentration should be reported in square brackets;
- The value of the lifetime is the result of a biexponential fit, as inferred from Fig. S2. Thus, the reported value is an average and it should be clearly stated;
- Add error bars in Figure 8; In equation 3, S should be replaced by sigma;
Author Response
Thank you very much for your detailed and an objective review of our work and appreciated the novelty of the manuscript.
1) English language was checked.
2) The method section was rewritten bearing in mind your comments.
3) This mistake was corrected.
4) Detailed spectrum description was added.
5) In Table 2 mistakes were corrected and relevant literature references were added.
6) There are several reason for using micellar solutions. Well known that one of the factors of increasing the intensity of the analytical signal in fluorescence analysis is the use of micelles and microemulsions. On the other hand the complex Tb(Phen) is limitedly soluble in water. Therefore, to improve the solubility of complex and increase the stability and analytical signal, we used micelle solutions.
The corresponding comment has been added to the manuscript.
7) In Table 1 the column with d values with errors was added.
8) In Figure 3 data for micelles of individual surfactants added.
9) According your comment the results were rewritten.
10) We totally agree with the note. Additional experiments are needed to evaluate the effectiveness of solubilization.
11) The description of calculation of fluorescence efficiency was added.
12) In Table 2 the errors were added. The decay curves were exponential fitted.
13) Lifetime of the Tb-phen-AA complex was added in Table 2 and described.
14) This mistake was corrected.
15) Figure 6 was correct
16) The Stern-Volmer plot was performed at λ = 544 nm.
17) This mistake was corrected.
18) The comment was added.
19) Equation 3 was corrected. In Fig. 8 error bars we did not add, since the curve was additionally approximated. We additionally added data of AA detection in tablets with errors.
Reviewer 4 Report
Manuscript Revision chemosensors-1220586
The manuscript "Terbium(III) as a fluorescent microheterogeneous probe for molecular detection of ascorbic acid" by Selivanova et al. describes the preparation, validation, and application of terbium (III) chelate, complexed with 1,10-phenanthroline and ascorbic acid (AA) as ligands, for the detection and quantification of ascorbic acid (AA) via quantitative fluorescence spectroscopy with a limit of detection of about 70 µM AA. The authors investigated the complexation process and studied the probe's quenching by AA via spectroscopic measures. They found that the kinetics of the photophysical quenching suffice the Stern-Volmer-Equation. Moreover, the effect of different surfactants on the solubilization efficiency was analyzed by determining the fluorescence decay of the probe complex. The optimal pH for the complex was found to be between 7 and 8. The authors state that the developed probe is sensitive, easy to prepare, and widely available, rendering an interesting sensor system to detect bioactive substances.
In general, the results presented in the manuscript are solid and might be of interest to the community in a broad context.
The introduction section is somewhat repetitive and bloated as it contains redundancies. The introduction section should be carefully revised and shortened. Moreover, some statements made are very vague and not entirely true – for example, the compound to be detected is not directly chemically changed by complexation but bound to the ligands – these molecular "changes" also entail changes in the physicochemical properties of the analyte. The advantages and disadvantages of the complexation probes should be discussed more comprehensively and less biased.
The most critical concern of the manuscript is some missing control experiments. Is the pH change induced by the addition of AA the main quenching factor? Is the probe selective towards AA? These questions should be experimentally addressed and clarified.
The manuscript is well written, and the language used is clear. It should be easy for international readers to follow and reproduce the described methods. Nevertheless, it is recommended to use one of the standard writing assistant software tools to eliminate minor language issues and further increase the clarity of language. The visual quality/quality of the presentation of the manuscript is good.
I recommend accepting the manuscript in Chemosensors after a major revision. Control experiments are needed to confirm the selectivity of the developed probe towards AA.
Specific comments:
- Please specify all abbreviations used in the manuscript (P1, L35: What does "NP" mean?)
- P2, L67: Why has the given LOD a negative LOD?
- P2, L66-L67: Why are the LOD's of the discussed probes so drastically different?
- Could you please explain how you determine fluorescence efficiency in the manuscript?
- P7, Figure 6: How does the system respond to changes in pH, but not concentration?
Author Response
Thank you very much for your detailed and an objective review of our work and appreciated the novelty of the manuscript.
1)The Introduction section was revised and shortened.
2) The details regarding some experiments were added. Particle size distribution curves of individual surfactants (Fig. 3), luminescent characteristics of the Tb(Phen)-AA complex in water (Table 2) were added into manuscript. In the Experimental part the details regarding the use of the buffer for building the calibration curve were added. Determination of ascorbic acid in tablets was also included.
3) All abbreviations were specified
4) The mistake was corrected.
5) The detection limit found by the sensitized fluorescence method depends on a number of factors. The main source of the formation of the analytical signal is the fluorescence of the terbium complex. It depends on the structure of the coordination sphere of the terbium ion, effective energy transfer from the ligand to the ion, pH of the medium, and solubilization in microheterogeneous media. Therefore, the value of the detection limit for different complexes of terbium and media can be very different.
In general, estimates of the detection limits by error blank experience is very conditional, therefore it follows clearly realize that their comparison is possible only however, with the same methods of calculating these estimates.
6) The ratio of the intensities of the transitions I(5D4−7F5)/I(5D4−7F6) may be a criterion of the luminescence monochromaticity of terbium (III) complexes. We used this ratio, estimated as the ratio of the intensities of the main energy transitions η = I(5D4−7F5)/I(5D4−7F6) at λ = 544 nm and λ = 489 nm, as the fluorescence efficiency of the complex.
7) To construct a calibration graph the buffer solution pH 7.0 was used. The description was added into manuscript.
Round 2
Reviewer 2 Report
The reviewer is satisfied with the response and the revised version.
Reviewer 3 Report
The Authors addressed the questions raised by this reviewer. The manuscript can be accepted for publication.
Reviewer 4 Report
Thank you for overhauling the manuscript. The quality of the paper is now much better and justifies publication in "Chemosensors".